# Dependencies of Surface Condensation on the Wettability and Nanostructure Size Differences

**DOI:** 10.3390/nano10091831

**Published:** 2020-09-14

**Authors:** Ming-Jun Liao, Li-Qiang Duan

**Affiliations:** School of Energy, Power and Mechanical Engineering, National Thermal Power Engineering & Technology Research Center Key Laboratory of Power Station Energy Transfer Conversion and System (North China Electric Power University), North China Electric Power University, Ministry of Education, Beijing 102206, China; liaomj666@gmail.com

**Keywords:** condensation, hydrophilic surface, hydrophobic surface, nanostructure size

## Abstract

When changing surface wettability and nanostructure size, condensation behavior displays distinct features. In this work, we investigated evaporation on a flat hydrophilic surface and condensation on both hydrophilic and hydrophobic nanostructured surfaces at the nanoscale using molecular dynamics simulations. The simulation results on hydrophilic surfaces indicated that larger groove widths and heights produced more liquid argon atoms, a quicker temperature response, and slower potential energy decline. These three characteristics closely relate to condensation areas or rates, which are determined by groove width and height. For condensation heat transfer, when the groove width was small, the change of groove height had little effect, while change of groove height caused a significant variation in the heat flux with a large groove width. When the cold wall was hydrophobic, the groove height became a significant impact factor, which caused no vapor atoms to condense in the groove with a larger height. The potential energy decreased with the increase of the groove height, which demonstrates a completely opposing trend when compared with hydrophilic surfaces.

## 1. Introduction

Both evaporation and condensation are common phase-change phenomena in daily life [1]. They are extensively used in various industrial fields, such as thermal management [2,3,4,5,6], heat pipes [7,8,9], cooling water harvesting [10,11,12], and electrical devices [13,14]. Changing the surface’s wettability is important in these applications, because different applications need distinct wettability surfaces. With the development of ultrafine manufacturing technology, the size of microelectronic devices, such as the integrated circuits inside cellphones and laptops, are becoming smaller and smaller. Therefore, evaporation and condensation phenomena at the micronano scale have a large number of applications in many aspects, and have become a widespread concern. Studying the heat transfer mechanism at the nanometer scale can help with the rapid miniaturization of electronic equipment.

Molecular dynamics (MD) as a numerical calculation method can exhibit the evolution of the interaction between atoms in the system over time at the nanoscale. Groups have conducted studies regarding heat transfer involving evaporation and condensation at the micronano scale. For example, Nagayama et al., [15] studied the effect of hydrophilic rectangular nanochannels on evaporation rate. They found that the presence of nanostructures reduced the surface thermal resistance and promoted the increase of the evaporation rate of ultrathin liquid film. Gao [16] quantitatively recorded the condensation process of water vapor on flat and pillar-structured surfaces. Further studies regarding the effect of the wettability on condensation and the phenomenon of heat transfer are needed. Hasan [17] investigated the effects of wettability (hydrophilicity and hydrophobicity) and surface materials on the evaporation of thin liquids on flat plates. The results showed that the surface of Al was the least effective in conducting heat, and Pt and Ag showed similar characteristics in both wettability conditions. The hydrophilic surface was more conducive to heat transfer. Ou [18] simulated the condensation of water vapor molecules on a mica substrate. Due to the inhomogeneity of the mica surface, the condensed liquid film was divided into two distinct adlayers, which illustrated that different wettability surfaces caused significant differences in condensation. Niu [19] studied condensation heat transfer on flat surfaces with different wettabilities, and the results proved that the condensation efficiency of a hydrophilic surface was higher than that of a hydrophobic surface. Both [18,19] focused their attention on the influence of the wettability, and ignored the effect of the microstructure. Hiratsuka [20] included three condensation modes by changing the interface wettability and microstructure height, i.e., drop, film, and discharge. Wang [21] demonstrated the combined effects of electric field strength and wettability on the condensation process. They found that the condensation rate decreased with the increase of the electric field strength and the condensation heat transfer was enhanced when the surface was more hydrophilic. Therefore, MD simulations have received widespread attention, as they are regarded as a valid method to solve the problem of heat transfer at the nanoscale.

All these papers refer to studies of either only the process of evaporation, or only the process of condensation. However, many applications attempt to combine these two processes together, such as the cooling of microelectronics [22], water desalination [23], and medical therapy [24,25]. Thus, triple-phase systems, which contain both evaporation and condensation, have received increasing attention. Recently, studies focused on a process with rapid boiling and condensing or evaporating and condensing processes. Yu [26] used MD to simulate the evaporation and condensation process of liquid argon in a cuboid composed of two platinum surfaces. Li [27] investigated evaporation and explosive boiling on a hydrophilic surface at two temperatures (130 and 300 K). Evaporation is more conducive to heat transfer, but explosive boiling will result in less heat for the phase change heat transfer. In addition, nanostructures promote condensation heat transfer, and the width of cuboid nanostructures will affect the condensation heat transfer. There exists a certain width to achieve the best performance of the condensation process. Yi [28] simulated the evaporation process of liquid argon film on two temperature surfaces (150 and 300 K) and the condensation process after the wall temperature dropped. They proved that the microevaporation process of the liquid film was consistent with that in the macroprocess. Kuri [29] simulated evaporation and condensation in a confined nanospace bound by a nanostructure in three different configurations for two different wall superheats, which were set at 110 and 250 K. They compared the heat transportation from the hot wall to the cold wall at these two heating temperatures with three nanostructured surfaces, and they demonstrated that the heat transportation was less in the case of explosive boiling (250 K) compared with normal evaporation (110 K). Liang [30] verified the Hertz–Knudsen Schrage equation by simulating the evaporation and condensation of liquid argon on the surface of platinum. All these papers in this paragraph focused on the dynamic movement of molecules and the influence of the heating temperature, and did not discuss the effect of wettability on condensation.

Despite suggestions about heat transfer that were proposed in previous articles, an indepth study of the dynamic and thermal mechanisms regarding evaporation and condensation at the nanoscale is still lacking. Therefore, in this paper, by performing molecular dynamic simulation, we investigated evaporation on a flat hydrophilic surface and condensation on both hydrophilic and hydrophobic nanostructured surfaces at the nanoscale. The work is organized as follows: First, the simulation model and computational details are presented. Then, the effects of nanostructure size on hydrophilic and hydrophobic surfaces condensation are revealed. The results are further analyzed by snapshots of the atom trajectories, the number of liquid atoms, temperature, potential energy, and heat flux. Finally, the findings obtained from this work are summarized.

## 2. Simulation Method

The molecular dynamics (MD) simulations were performed using the largescale atomic/molecular massively parallel simulator (LAMMPS) package [31] to build a simulation system. The software of OVITO (GmbH, Germany) was used to display the visual graphs of the system and the atomic motion and to analyze the MD simulation results. As shown in Figure 1 with perspective and front views, a solid flat plate made of copper was placed at the bottom of the system, which was the heat source. Argon as a working fluid was divided into two states of liquid and vapor and placed on the bottom plate and middle region, respectively. At the top of the system was a nanostructured plate also made of copper, and this was the condensing wall. Face centered cubic (FCC) atoms were adopted to model the walls at two ends and the liquid/vapor working fluid with a lattice of 3.615, 5.744, and 36.251 Å. A periodic boundary condition was adopted in the x and y directions with sizes of 7.35 × 7.35 nm, whereas a rebound condition was applied in the z direction of 62.2 nm, which ensured that the total energy of the system did not change.

For the copper walls at the two ends of the system, the outermost two layers were defined as the fixed wall, to prevent the atoms flying away and plate deformation, and the remaining layers were treated as a thermostat with a heating temperature of 130 K and a cooling temperature of 85 K using the Nosé−Hoover thermostat [32,33]. For the liquid argon atoms, the film thickness was 3.6 nm with the number of 4739 atoms. For the dispersed vapor argons, they filled between the liquid film and the upper wall, with the number of 338 atoms. When the liquid layer was heated by the heat source, evaporation was triggered, and new vapor argons moved to the middle region to mix with the original ones. Finally, when the vapor argons arrived at the cold wall, condensation occurred. Here, the nanostructure grooves for vapor condensation had three different widths in the x-direction at 1.8, 2.4, and 3.2 nm, and for every width, three different heights of 0.8, 1.8, and 2.8 nm in the z-direction were constructed.

The interactions between Cu–Cu, Ar–Ar, and Cu–Ar are all described by Lennard-Jones 12−6 potentials with a cutoff distance of 11.9 Å, expressed as
(1)$$V_{ij}=4\varepsilon \left[{\left(\sigma /r\right)}^{12}-{\left(\sigma /r\right)}^{6}\right]$$
where *r*_ij_ is the distance between particles *i* and *j* (Å), *ε* is the minimum potential value (eV), and *σ* is the particle spacing when the potential is zero (Å). The method of adjusting ε to obtain the distinctive wettabilities is widely adopted in MD simulations [34,35,36,37,38]. Therefore, in this work, two kinds of wettability for the condensation were studied. When ε_Cu-Ar_ is 0.0065 eV, it represents the hydrophilic surface, and when ε_Cu-Ar_ is 0.0025 eV, it represents the hydrophobic surface. In addition, the heated wall is hydrophilic with ε_Cu-Ar_ = 0.0065 eV.

After preparation of the initial configurations and interaction force fields, three consequent stages were performed. The first was the energy minimization, which used the conjugate gradient algorithm, with the potential energy and force stopping tolerance of 1.0 × 10^−5^ eV and 1.0 × 10^−6^ eV/Å, respectively, and the number of the steps was 10,000. Then, the system was equilibrated in a constant number of atoms, volume, and temperature (NVT) ensemble at 85 K for 0.8 ns. At the end of this stage, the copper plate and argon atoms were all stable at the uniform temperature, and their energy maintained a constant value. Therefore, during this stage, the system reached the steady state, preparing for evaporation and condensation in the next step. Thus, in the third stage, the temperature of the bottom copper plate suddenly improved to 130 K to heat the liquid argon film. A constant number of atoms, volume, and energy (NVE) ensemble was applied to the liquid and vapor argon atoms to ensure that they only absorbed energy from the heated wall and released energy to the cold wall. The upper nanostructured plate was the same as the second stage with a constant temperature of 85 K. In the third stage, the system was run for 5 ns. Throughout the above three stages of simulations, the time step was set to the same value of 1 fs, and the velocity Verlet algorithm was used to solve the Newtonian motion equations for each atom. The positions and velocities used in analysis in this work were stored and calculated every 1000 time steps.

## 3. Results and Discussion

### 3.1. Effect of the Nanostructure Size on Hydrophilic Surface Condensation

Figure 2 exhibits the trajectories of the atoms for the whole system, where the bottom flat plate was set as the heat source, and the nanostructured plate with the groove width (w) of 1.8 nm and the height (h) of 0.8 and 2.8 nm was set as the cold source. As the temperature of the bottom plate improved to 130 K, the temperature difference between the copper and liquid argon atoms induced evaporation, which made new vapor argon atoms move to the upper position. Due to the thickness of the liquid film at 3.6 nm, 130 K could not reach its explosive boiling temperature. Thus, only the phenomenon of evaporation was observed, which was consistent with the conclusion in our previous paper [39]. The new heated vapor atoms with higher energy were more active, resulting in drastic collisions in the middle region. With the proceeding of evaporation, most liquid atoms were heated up. These active and high energy atoms continuously transferred energy through the collision, promoting vapor atoms in the middle region to move to the top plate. When they arrived at the nanostructured plate, they released the energy to the cold wall and finally condensed on it. Therefore, from the snapshots of Figure 2, we found that the liquid film gradually moved from the bottom plate to the top plate. As the two plates were hydrophilic, there was always a layer of argon atoms on the top plate at 0 ns and bottom plate at 5 ns. The condensation was simultaneously observed in the groove and the flat areas.

From the snapshots of Figure 2, it was difficult to determine the effect of nanostructure size on condensation. To quantitatively illustrate the influence of the distinctive groove heights and widths, the numbers of liquid argon atoms (N) were computed, as shown in Figure 3. For all the situations, the liquid atoms dramatically decreased before 2 ns, and then slowly increased in the remaining time. That was because, in the initial stage (<2 ns), evaporation was the dominant factor, which resulted in the phase transition from liquid atoms contacting the heated wall, while, with the vapor atoms moving towards the cold wall, condensation became the dominant factor in the later stage (>2 ns). Therefore, more vapor atoms condensed and even exceeded the number of vapor atoms from evaporation. On the other hand, due to the different dominant factors in these two stages, the influence of nanostructure size in the initial stage was negligible. Thus, although the groove heights were different, the numbers of liquid atoms were nearly the same. However, for the condensation dominant part, the number of liquid atoms had an obvious discrepancy. For these three groove widths, the number of liquid atoms with higher groove height (h = 2.8 nm) was always larger than that with a lower groove height (h = 0.8 nm). This is because a nanostructure with a higher groove height will have more condensation areas, and thus faster condensation leads to more liquid atoms. With the increase of groove width, we found that the differences between them were marginal. This indicated that width had little effect on the number of liquid atoms compared to height. Thus, to achieve a more obvious condensation effect, adjusting the height of the nanostructure was more efficient than adjusting the width. The number of liquid atoms affected the temperature of the argon atoms. Therefore, the effect of nanostructure size on temperature of all the argon atoms is discussed in the next part.

By comparing both Figure 3 and Figure 4, the trends of the curves were exactly opposite, which indicated that a larger number of liquid atoms induced a lower temperature, and, conversely, a smaller number of liquid atoms caused a higher temperature. In fact, the variation of temperature was attributed to the processes of both evaporation and condensation. When the heated source with 130 K suddenly acted on the liquid film placed on it, the high temperature brought about the evaporation and the argon atoms were heated up immediately. Thus, the temperature dramatically increased in the initial stage. As the evaporated argon atoms moved towards to the cold wall and condensed on it, the cooling effect came in to play, which made the temperature slightly decrease in the later stage. The reason for these two stages having distinct rates of change was that only evaporation occurred in the first stage, while both condensation and evaporation took place in the second stage. At the final state, with most vapor atoms finishing the condensation, the temperature was nearly equal to the initial temperature. For the influence of the groove height, a larger height brought more condensation area, which led to faster condensation, finally inducing a significant temperature decline. From Figure 4, we found that a larger groove height (h = 2.8 nm) always contained the fastest decline and lowest temperature. The smaller groove height (h = 0.8 nm) exhibited the slowest decline. On the other hand, similar to the number of liquid atoms, the groove width difference also had little effect on the temperature (Figure 4).

For further investigation of the effect of nanostructures on condensation, we calculated the potential energy (E) between argon atoms and the cold wall (Figure 5). Fundamentally speaking, the vapor atoms condensed on the cold wall were determined by the attractive potential energy between the atoms and the substrate. The situation of condensation and the liquid atom numbers on nanostructured surfaces were influenced by the potential energy. Therefore, the values of the potential energy and its changes were the basic reasons for the variations of the other indicators (the number of liquid atoms, the temperature, and the heat flux). In particular, for studies on the nanoscale, the interaction between molecules becomes more significant. The specific phenomenon, processes, and results are all dominated by potential energy. Therefore, the effect of the potential energy is significant. In general, potential energy shows a trend of gradually increasing and then remaining basically unchanged over time, no matter what the geometric parameters of the nanostructure are. That is because the vapor atoms gradually arrive at a cold wall and condense on it, inducing the number of absorbed atoms to be larger. Thus, the attractive potential increases with the more liquid atoms on the cold nanostructured plate. When the groove and flat surfaces are both covered by argon atoms, the potential energy will not continue to increase as before. Even if the vapor atoms sequentially condense, because the distance between the condensed atoms and the cold wall is too large, the attractive potential is basically negligible. With the increase of the groove height, the time it takes to achieve full coverage of the surface also increases. Therefore, from the three curves in Figure 5, we found that the potential energy reached stability more slowly if the groove height was larger. Similarly, when the groove width increased, the time also increased. By comparing these three graphs, we made these conclusions.

Figure 6 demonstrates the condensation heat transfer on distinctive nanostructured surfaces by calculating the heat flux (H) at the region near the cold wall by 1 nm. As the vapor atoms with high temperature condensed on the cold wall to release heat, the heat flux began to decrease in the initial stage. When the statistical region was covered by argon liquid atoms, this area began to absorb heat through those atoms that continued to condense on the upper part of the liquid film. Thus, the heat flux exhibited an increasing trend in the second stage. For a small groove width (Figure 6a), only if the height was large enough (2.8 nm), did the heat flux show an obvious distinction compared with other two heights. With the increase of the groove width (Figure 6a,b), the difference between larger heights (1.8 and 2.8 nm) became smaller, otherwise, the distinction of the small height (0.8 nm) was dramatic. This phenomenon can be attributed to the competition of groove height and width, as to which is the dominant factor. For example, when the groove width was small (1.8 nm), the height did not affect the heat flux for smaller ones (0.8 and 1.8 nm), which indicates that the width was the dominant factor. However, when the groove width became larger, a slight change in height caused a significant variation in the heat flux, that is to say, the height became the dominant factor.

### 3.2. Effect of Nanostructure Size on Hydrophobic Surface Condensation

When the wettability of the cold wall was hydrophobic, the condensation behavior also changed. The snapshots of the atom trajectories for the whole system with nanostructure hydrophobic surfaces are shown in Figure 7. The initial stage was the same as the situation of hydrophilic surfaces, in which only evaporation occurred and the vapor atoms collided with each other in the middle region. However, when the vapor atoms arrived at the cold wall, a different phenomenon was exhibited. As seen in Figure 7, from 2 ns, condensation can be observed. Differently to the simultaneous condensation in both the groove and flat areas for the hydrophilic nanostructure surface, condensation favored the flat areas for the hydrophobic one. For small groove height (Figure 7a), once the condensed liquid film reached a certain thickness on the left and right flat areas, these two parts merged into one through the middle groove. Although the condensation process was different for the hydrophilic surface, the final state was similar. However, for the large groove height (Figure 7b), the condensation only proceeded on the flat areas. There were no condensed liquid atoms in the groove until the end of the condensation process. This is clearly distinct to the situation of the hydrophilic nanostructured surface. Therefore, it is necessary to conduct research regarding condensation on hydrophobic surfaces.

According to Figure 3 and Figure 8, we found that the overall trends of the number of liquid atoms for both hydrophilic and hydrophobic surfaces were similar. One of the distinctions was that the final numbers of liquid atoms for hydrophobic surfaces were all smaller than those for hydrophilic surfaces, at approximately 1600 vs. 2100. One reason is that the adsorption capacity of argon atoms for the hydrophobic surface is weaker, which will be further discussed in the next part (the analysis of the potential energy), and another reason is the lack of condensation in the groove areas. Both these factors contribute to the decrease in the number of liquid atoms. The difference between the small groove height (0.8 nm) and larger groove heights (1.8 and 2.8 nm) became more obvious with larger groove width (Figure 8b,c). This was because the groove areas had no condensation with the larger height but still condensed with the small height, which can be seen in Figure 7. Therefore, the groove height was a significant impact factor for hydrophobic surface condensation.

The variation of temperature (Figure 9) was also opposite to the number of liquid atoms, which was consistent with the conclusion drawn from the hydrophilic surface condensation. Thus, the primary cause for the change of temperature was again the competition between evaporation and condensation. However, with the different numbers of condensed liquid atoms with smaller and higher groove heights, more obvious temperature distinctions were shown in the first stage (<2 ns). On the other hand, as the hydrophobic surface had a weaker adsorption capacity of argon atoms, a slighter slope of the temperature drop was included in the second stage (>2 ns), which was compared to the temperature on hydrophilic surfaces (Figure 4). Therefore, the wettability had an important influence on the argon temperature for the whole system.

With the increase of the condensed liquid atoms, the potental energy gradually increased and then remained stable on the hydrophobic surface (Figure 10), which exhibited a similar trend to the situation of the hydrophilic surface (Figure 5). Surprisingly, through comparing these two figures, the potential energies showed completely opposite trends on two wettability surfaces with the increase of groove height. To analyze the reason for this special phenomenon, the snapshots of the evaporation and condensation process for the whole system were studied carefully. Finally, we found that for the small groove height (h = 0.8 nm) with the three groove widths, the condensed argon atoms covered the grooves and flat areas, which indicated that more atoms were absorbed on the cold wall, inducing a higher attractive potential. With the increase of the groove height, the condensed atoms in the groove areas were decreased, and no condensation was observed for the height of 2.8 nm. Therefore, the potential energy decreased with the increase of the groove height.

## 4. Conclusions

We investigated evaporation on a flat hydrophilic surface and condensation on both hydrophilic and hydrophobic nanostructured surfaces at the nanoscale using a molecular dynamics simulation. Copper was chosen as the heat and cold source. Liquid and vapor argon atoms as the media of heat transfer were used to fill between these two walls. Eighteen types of cases (including nine on two wettability surfaces) were designed on the cold wall to study the influence of wettability and nanostructure size on condensation. The snapshots of the system, number of liquid atoms, argon temperature, potential energy, and heat flux were exhibited and analyzed.

The simulation results on hydrophilic surfaces indicated that the whole process could be divided into two stages with 2 ns as the boundary. Throughout the research period, both the evaporation and condensation processes affected the number of liquid atoms and argon temperature, and determining which process had become the dominant factor was crucial to the system. The larger groove width and height increased the condensation area, inducing more liquid atoms and a quicker temperature response. With the increase of the groove height and width, the potential energy reached stability more slowly. For the condensation heat transfer, when the groove width was small, the change of groove height had little effect, while still causing a significant variation in the heat flux with a large groove width. Compared to previous similar studies, this was the first study to combine the variation of height and width of the nanostructure to investigate the effect on condensation heat transfer. For the hydrophilic nanostructured surface, both the height and width affected the condensation heat transfer.

If the cold wall became hydrophobic, the condensation behavior also changed. One of the strongest distinctions was that the groove height became a significant impact factor, which caused no condensation with a larger height. As the adsorption capacity of argon atoms for a hydrophobic surface is weaker, this also provides new findings and conclusions. The final numbers of liquid atoms for hydrophobic surfaces were all smaller than those for hydrophilic surfaces. The difference between the small groove height and larger groove height became more obvious with a larger groove width. More clear temperature distinctions were shown in the first stage (<2 ns), and a slighter slope of temperature drop occurred in the second stage (>2 ns), when compared to the temperatures with hydrophilic surfaces. The potential energies decreased with the increase of the groove height, which showed completely opposing trends compared with those on hydrophilic surfaces. As far as we know, there are no previous studies on condensation on hydrophobic nanostructured surfaces. Therefore, this investigation has significant potential for industrial applications. To better apply this research in practice, we will expand the surface from a single groove to a series of microstructures in our future work.

## Figures and Tables

**Figure 1 nanomaterials-10-01831-f001:**
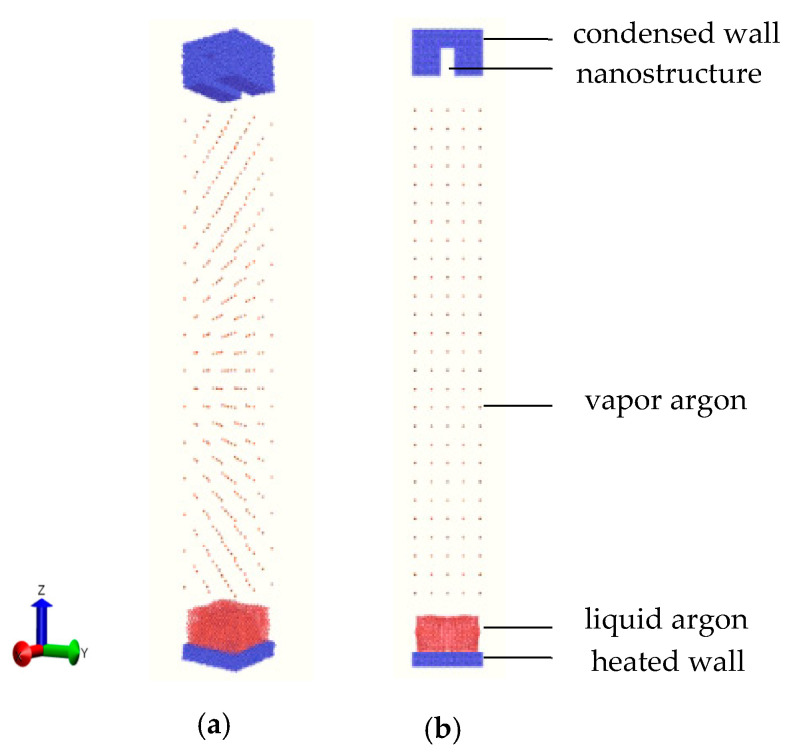
Schematic diagram of the (**a**) perspective and (**b**) orthographic views of the simulation domain.

**Figure 2 nanomaterials-10-01831-f002:**
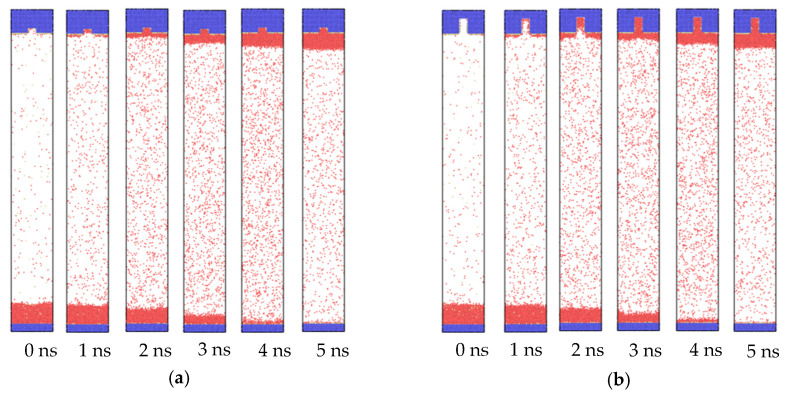
Snapshots of the atoms’ trajectories for nanostructures with a groove width of 1.8 nm and height of (**a**) 0.8 nm and (**b**) 2.8 nm on hydrophilic surfaces.

**Figure 3 nanomaterials-10-01831-f003:**
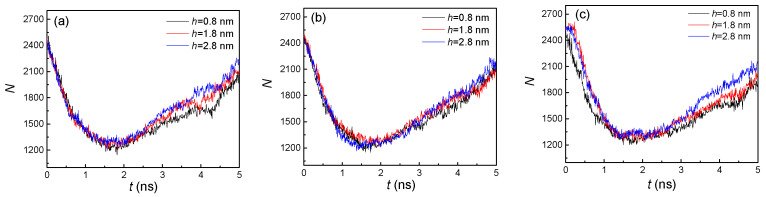
The statistical number of liquid argon atoms with three groove heights; from left to right, the groove widths are (**a**) 1.8, (**b**) 2.4, and (**c**) 3.2 nm on hydrophilic surfaces.

**Figure 4 nanomaterials-10-01831-f004:**
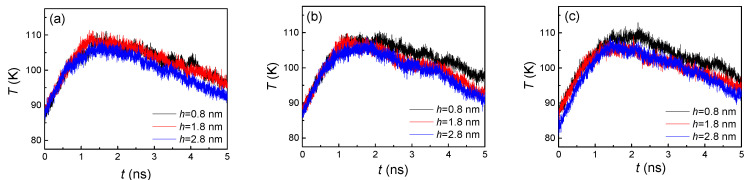
The temperature of argon atoms with three groove heights; from left to right, the groove widths are (**a**) 1.8, (**b**) 2.4, and (**c**) 3.2 nm on hydrophilic surfaces.

**Figure 5 nanomaterials-10-01831-f005:**
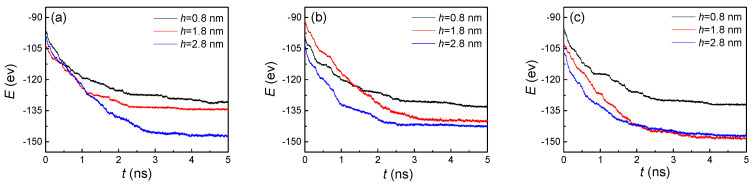
The potential energy between argon atoms and the cold wall with three groove heights and widths of (**a**) 1.8, (**b**) 2.4, and (**c**) 3.2 nm on hydrophilic surfaces.

**Figure 6 nanomaterials-10-01831-f006:**
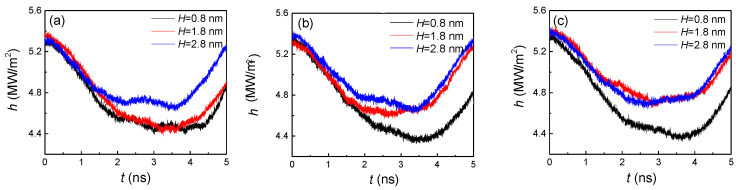
The heat flux at the region near the cold wall by 1 nm with three groove heights and widths of (**a**) 1.8, (**b**) 2.4, and (**c**) 3.2 nm on hydrophilic surfaces.

**Figure 7 nanomaterials-10-01831-f007:**
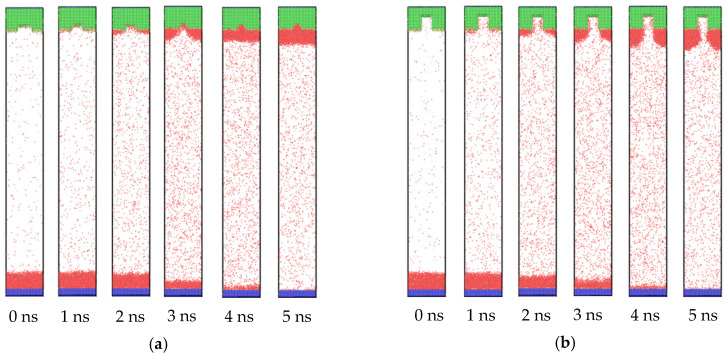
Snapshots of the atom trajectories for nanostructures with a groove width of 2.4 nm and heights of (**a**) 0.8 and (**b**) 2.8 nm on hydrophobic surfaces.

**Figure 8 nanomaterials-10-01831-f008:**
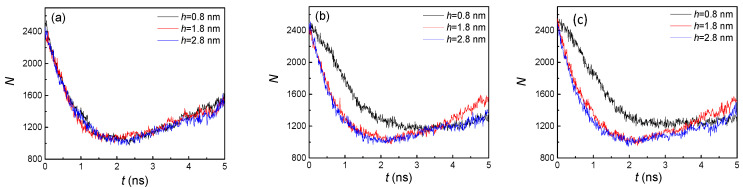
The statistical number of liquid argon atoms with three groove heights and widths of (**a**) 1.8, (**b**) 2.4, and (**c**) 3.2 nm on hydrophobic surfaces.

**Figure 9 nanomaterials-10-01831-f009:**
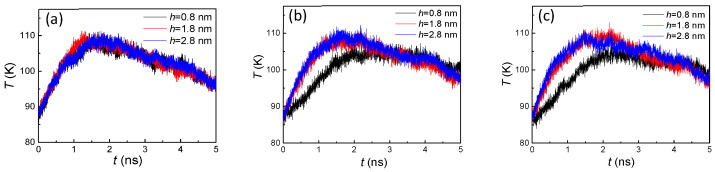
The temperature of argon atoms with three groove heights and widths of (**a**) 1.8, (**b**) 2.4, and (**c**) 3.2 nm on hydrophobic surfaces.

**Figure 10 nanomaterials-10-01831-f010:**
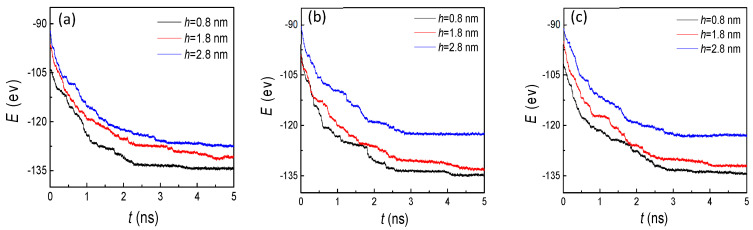
The potential energy between argon atoms and the cold wall with three groove heights and widths of (**a**) 1.8, (**b**) 2.4, and (**c**) 3.2 nm on hydrophobic surfaces.

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
