# Peer review of "Dependencies of Surface Condensation on the Wettability and Nanostructure Size Differences"

_nanomaterials, 2020, doi:10.3390/nano10091831_

Round 1
Reviewer 1 Report
no comments
Author Response
Dear reviewer,
Thank you very much.
Best regards,
Liqiang Duan
Reviewer 2 Report
There are some serious issues that must be dealt with before this can be publishable. The first major issue is the grammar and incomplete sentences, almost all of the sentence's grammar need to be corrected as I still cannot fully understand on what some sentences even mean. I understand that it may be difficult for non-english speakers to translate their work into another language, however the quality of English must reach a satisfactory level if one seeks to publish an article. In addition the literature review will need to be improved as there lies a lack of comprehension of the authors work. The details of such improvements shall be highlighted at the end.
Another issue is, how is the surface wettability defined at a molecular scale? I would think at a molecular level surface wettability is defined from the change in surface energy, however this is not said or explained in any of the sentences. Moreover this could have been used to explain the absorption capacity of argon atoms for hydrophobic and hydrophilic surfaces. Following on, the setup of the simulation should be verified or calibrated with prior work, to determine the validity of the simulation results. It would also be useful to state some of the equations that the simulation uses to determine the temperature and heat flux of the argon atoms and surface. This will help the readers to understand the trend of the results displayed for each of these factors.
Subsequently the purpose or significance of this paper from a practical view point is questionable. Why is the wettability of the surface and the dimensions of the nanostructure cavity are both accessed independently? Many mechanical engineers have shown that these two factors to be mutually inclusive, unless the surface is chemically modified to prohibit such atoms. However this was not stated or cited within the literature imposed. Plus within the introduction it is not stated about the importance of changing the surface's wettability within the applications that are discussed. To further point out, why is argon chosen as the working fluid in this study? Is argon widely used within the field of condensation-evaporation systems?
P1, L9: A dot here is not needed.
P1, L10: What is the structure? Nanowires? Nanopillars?
P1, L11: Sentence should be re-written, as there are some grammar errors.
P1, L12: Is it not at a molecular level?
P1, L14: Make? This is a very contradicting statement, according to the 1st law of thermodynamics.
P1, L14: What does "quicker temperature response" mean?
P1, L15: Items? These are not items.
P1, L16: "condensation areas"? Or is it where condensation occurs?
P1, L17: How is "small" defined in this context?
P1, L18: Sentence is not complete.
P1, L20: "no condensation"? This could be misinterpreted from it affecting the ambient condensation. Specifically in this context it affects how wet the surface will become (due to condensation) or in this case, the number of liquid atoms.
P1, L20: "energies"? You are just talking about one type of energy.
P1, L28: Where is reference 6?
P1, L28: Grammar check. And what electrical devices?
P1, L29: Cellphones and laptops are not micro-electronic devices.
P1, L35: Check grammar.
P1, L36: This part of the sentence does not belong here. It would make more sense for it to be in the last sentence. And the grammar needs to be corrected.
P1, L38: It would be better to say et al. to be more formal.
P1, L42: Check grammar. The sentence does not make sense. What did Gao conclude from the recordings?
P1, L44: By now it is not defined what wettability means in context with this paper.
P1, L44: Was it "effects of wettability on different surface materials"? And what were the surface materials?
P2, L45: Say that it is aluminum. Also I fail to see how aluminum is not effective in conducting heat as it has a high thermal conductivity. Was it an alloy? In addition why wasn't the heat transfer performance of Ag (gold) and Pt (platinum) mentioned alongside?
P2, L46: In what aspect is it more conductive? If hydrophilic surfaces promote surface wetting, wouldn't the wet surface create a thermal resistance?
P2, L49: Check grammar, Also did the inhomogeneity of the surface create different wettability characteristics?
P2, L51: What were the different wettabilites?
P2, L52: Higher than what? Also how does a hydrophilic surface promote more heat transfer?
P2, L53: "obtained" is not the right word to use here.
P2, L54: What is "interface wettability"?
P2, L55: What is "discharge condensation"?
P2, L57: Again, how did the condensation heat transfer become enhanced when the surface was more hydrophilic? Many literature papers show that they do not.
P2, L60: Please give some examples of these applications.
P2, L62: Check grammar.
P2, L66: What does "heat of phase transformation" mean?
P2, L67: Did Li use "nanostructures" in the investigation?
P2, L68: What width? The nanostructure cavity width?
P2, L69: Effect? or performance? And this sentence is not complete.
P2, L69: What were the "two kinds of temperature surfaces"?
P2, L71: Consistent to what? The evaporation rate?
P2, L72: Why is there a capital N in "Nano".
P2, L72: "by a"
P2, L75: Check grammar.
P2, L76: How was it verified?
P2, L76: How was the Hertz Knudsen Schrage equation verified?
P2, L79: All the papers? Or all the papers in this paragraph?
P2, L83: Check the grammar in the entire sentence.
P2, L87: "snapshots" is not a formal word to use.
P2, L88: "Liquid atom numbers"? Or the number of liquid atoms?
P2, L92: This is not an experimental system.
P2, L93: What is OVITO?
P2, L95: "made by copper atoms"?
P3, L96: Argon, being the working fluid of this simulation should have been mentioned in the abstract.
P3, L98: Why not just say "made by copper"?
P3, L98: Why are specifically (FCC) atoms used?
P3, L101: Space is needed between "x7".
P3, L101: What does "rebound condition" mean?
P3, L101: Was this at the same x and y position?
P3, L104: The vapour argon atoms can hardly be seen. Also why are they the same colour as the liquid argon atoms? This will be very difficult to create a distinction between the two phases.
P3, L106: Part of sentence poorly written.
P2, L106: What "other layers"?
P3, L108: "argons"?
P3, L112: Check grammar.
P2, L113: It is not "nanostructures" This is just a simulation of a nanostructure cavity.
P3, L118: What is r and Vij? And what are the units for each parameter?
P3, L125: Is it potential energy?
P3, L125: Shouldn't the tolerance be plus or minus?
P4, L127: Check grammar.
P4, L128: What "energy"?
P4, L131: "increases". Plus the grammar should be checked in the sentence.
P4, L135 The sentence is not complete.
P4, L138: The grammar in the whole paragraph should be checked.
P4, L142: This should be mentioned in the paragraph where the boundary conditions of the model are being defined.
P4, L155: From Figure 2, there is hardly enough liquid atoms shown to quantify a liquid film.
P4, L156: The grammar should be checked in the paragraph.
P5, L163: What does "last time" mean?
P5, L164: Space is needed in"<2" .
P5, L167: "number of the evaporation"?
P5, L170: Was this inside the cavity?
P5, L173: Space is needed "h=".
P5, L175: Sentence is poorly written. And the last part of the sentence has nothing to do with the nanostructure groove height.
P5, L177: Sentence is poorly written again.
P5, L179: What "liquid atoms"?
P5, L179: The vapour or liquid argon atoms?
P5, L180: Temperature of what?
P5, L181: What does "the trends of the curves are exactly opposite" mean?
P5, L182: Reduces the temperature of the liquid argon atoms?
P5, L183: Is it fluctuations? And by now it was not mentioned that the temperature was fluctuating.
P5, L192: Again this statement has nothing to do with its influence on its geometry.
P5, L196: The entire paragraph should be checked for grammar issues.
P5, L198: Not needed to say.
P5, L198: Space needed in "2.8nm".
P5, L199: This would be better said on the individual figures themselves.
P6, L201: Is it "liquid argon atoms"?
P6, L204: The "(E)" should be after potential energy.
P6, L208: What other indicators?
P6, L210: Grammar needs to be corrected on these revised sentences.
P6, L211: Its discussion? Or its effect?
P6, L212: At what time does this transition occur?
P6, L223: The grammar should be corrected in this paragraph. And this last sentence is not complete.
P6, L228: Please use a different parameter to represent the heat flux, as (h) already represents the height of the nano-cavity. Usually heat flux is denoted by (H).
P6, L228: Was it adjacent to the wall?
P6, L230: What is the "statistical region"?
P7, L238: Smaller than what?
P7, L239: Why does this phenomenon occur?
P7, L246: Please stick to one tense in a sentence.
P7, L258: Why is it distinct to hydrophilic nanostructure surfaces?
P7, L259: Again check grammar in paragraph.
P8, L270: Is it in the next paragraph?
P8, L276: Grammar needs to be corrected in the paragraph as most sentences do not make sense.
P8, L277: What does opposite mean here?
P8, L282: How do you know this? Is there a theory that supports this?
P9, L296: "unnatural" is not a good term to use.
P9, L297: Sentence is not complete. It is not explained why this occurs.
P9, L298: For the hydrophobic or hydrophilic surface? Or both?
P9, L304: Was the solid heated wall hydrophilic?
P10, L311: Was it just on the hydrophilic surfaces?
P10, L336: Grammar should be checked in the conclusion.
Reviewer 3 Report
In my view, the revised version is fully appropriate for publication, the authors did improve the original manuscript, which required minor modification anyway.
Author Response
Dear reviewer,
We appreciate your positive comments. We have checked the grammar and content carefully. We have made some minor modifications and improved the manuscript.
Thank you.
Liqiang Duan.
This manuscript is a resubmission of an earlier submission. The following is a list of the peer review reports and author responses from that submission.
Round 1
Reviewer 1 Report
This is well-written manuscript describes the results of an appropriately designed research project. The work has been performed professionally, only a final English usage and grammar check should be performed. As far as science goes, in my view, it can be accepted for publication in Nanomaterials.
Reviewer 2 Report
The significance of this manuscript is insufficient. Generally speaking, the research quality is low. For example, the Introduction section is merely a result from simple copy-and-paste of references, without in-depth discussions. There are various small errors, such as plural-singular mistakes, throughout the entire manuscript. The output regarding the effect of the groove width (or even the height) has been largely overstated. Also, if the simulation is based on one single microgroove site, then the results are hardly applicable to real-world applications. Lastly, it makes little sense to discuss condensation on nanostructure, without taking into account of the capillary condensation.
L14, Why more liquid atoms? And a quicker temperature response, and slower potential energy decline. 

L38: Wrong author name. The first author of Ref 16 is not called Mohammad Gao. This is a careless error.
L41: Al is a commonly used engineering material. What makes the results that showed “the surface of Al is most favorable for heat transfer” new?
L42: What makes the hydrophilic surface “more conducive to heat transfer”?
L44: It is hardly surprising that surface inhomogeneity will cause condensation to show distinct results. What is the point of citing Ref 18?
L52: Reword the following sentence to make it clear: All these studies are presented the evaporation and condensation separately.
L53: What is a triple-phase system?
L64: The discussion of Kuri’s work is not clear. Reword the sentence.
In addition, it is without a doubt that surface area and the nanostructure height will affect the rate of condensation. But how?

L92: The distance between the top and bottom of Figure 1 is utterly long. Does it scale with the real physics?
L102-3: Why are the widths given the following values, 1.8, 2.4 and 3.2 nm? The range of these values is too small to reflect the effect of the width on the condensation and evaporation performances. Similarly, what is the physics behind the chosen heights of 0.8, 1.8 and 2.8 nm?
Is it mainly to make the simulation easy?
Also, why is the distance between the plates much larger than the widths and heights?
L107: “Where” is not meant to be capitalised.
L109: The first letter of “when” should be capitalised.
In addition, when combining two symbols into one term, it is common to make one of the symbols a subscript or superscript. This makes “εCu-Ar” strange.
L131: Are the simulation results of Figure 2 based on one micro-grove only? If that is the case, then it does not seem to be realistic.
L183: The captions for Figure 3 and Figure 4 are misleading. It appears that the sub-sections (a-c) correspond to three groove widths.
In addition, the difference from (a) to (c) in Figure 3 and Figure 4 is marginal. This has not been addressed clearly in the text. While the discussions are mainly around the effect of the groove height, the effect of the width is hardly touched upon.
L191: What does it mean that “the discussion about the potential energy is significant”? In fact, there is not much significance within the discussion. 

L291: What does it mean that “both the evaporation and condensation affect the number of liquid atoms and argon temperature”? Isn’t it expected that the number of liquid atoms will differ between evaporation and condensation?
In addition, it is misleading to refer to “which one”? Is it between evaporation and condensation?
Reviewer 3 Report
The manuscript by Ming-Jun Liao and Li-Qiang Duan focuses on the application of the computational molecular dynamic approach on the analysis of the chemical/physical changes on the surface of a given material.
In particulars, they investigate the evaporation on the flat hydrophilic surface and 11 condensations on both hydrophilic and hydrophobic nanostructured surfaces at the nanoscale are 12 investigated by the molecular dynamics simulation.
The manuscript is very interesting and the results well described and presented. I have only minor suggestions, that however might improve the manuscript and make it suitable for a widespread audience.
Minor points:
-The authors have to improve the method description.
-The authors have to improve the Conclusion section, comparing their results with those of other authors.